# The Effect of Protein Source on the Physicochemical, Nutritional Properties and Microstructure of High-Protein Bars Intended for Physically Active People

**DOI:** 10.3390/foods9101467

**Published:** 2020-10-15

**Authors:** Jan Małecki, Igor Tomasevic, Ilija Djekic, Bartosz G. Sołowiej

**Affiliations:** 1Department of Milk Technology and Hydrocolloids, Faculty of Food Sciences and Biotechnology, University of Life Sciences in Lublin, Skromna 8, 20-704 Lublin, Poland; j.malecki@eurohansa.com.pl; 2EUROHANSA Sp. z o.o., ul. Letnia 10-14, 87-100 Toruń, Plant in Puławy, ul. Wiślana 8, 24-100 Puławy, Poland; 3Department of Animal Source Food Technology, Faculty of Agriculture, University of Belgrade, Nemanjina 6, 11080 Belgrade, Serbia; tbigor@agrif.bg.ac.rs; 4Department of Management of Food Safety and Quality, Faculty of Agriculture, University of Belgrade, Nemanjina 6, 11080 Belgrade, Serbia; idjekic@agrif.bg.ac.rs

**Keywords:** animal and plant proteins, computer vision system, nutritional value, texture, water activity, viscosity, microstructure, heavy metals, amino acids

## Abstract

The purpose of this study was to investigate the effect of protein sources (algae, pumpkin, wheat, sunflower, rice, soy, hemp, pea, and whey) on selected physicochemical, nutritional, and structural parameters of high-protein bars. Texture properties, such as hardness, fracturability, cohesiveness, and adhesiveness, have changed depending on the type of protein used. A significant increase, in particular the hardness parameter relating to the control sample (whey protein concentrate—WPC80), was noted for bars containing algae, sunflower, and wheat proteins, with high values of the adhesiveness parameter concurrently. The use of proteins from algae, pea, and wheat resulted in a significant reduction in the water activity of the finished product compared to WPC80. Bars made with the use of wheat, hemp and pumpkin proteins had noticeably higher viscosities than other samples. Color of the tested bars measured by means of Computer Vision System (CVS) was from light cream (soy, pea) to dark green (hemp, pumpkin). Bars prepared of wheat and algae proteins had the highest nutritional value, while the lowest one was recorded in products containing sunflower and hemp proteins. There was a clear differentiation of amino acids (g/100 g) and microstructure in bars depending on the type of protein used. However, a slight similarity can be found between whey and soy proteins (amino acids) and between whey and sunflower proteins (microstructure). Obtained results suggest that selection of the right type of protein for a given application may have a significant impact on the physicochemical features and microstructure of high-protein bars and their nutritional values.

## 1. Introduction

High-protein products, including bars, have recently become extremely popular. In particular, since products enriched in protein or in which protein is the main ingredient, can be used in products intended for a wide group of consumers [1].

This type of product can be used in the segment of quick snacks (designed to temporarily satisfy hunger), in sports nutrition (muscle tissue growth) or products intended for nutrition of the elderly and sick people who are at risk of developing sarcopenia [2]. As a result of such a large interest in high-protein products on the market, manufacturers meet the consumers’ requirements and constantly develop recipes for innovative products that can be part of current trends in healthy and functional nutrition [3]. For this purpose, manufacturers searching for suitable alternatives to commonly used ingredients, such as high fructose and glucose syrups, fats or allergenic proteins, into their alternative components, e.g., polyols, fructo-oligosaccharides, or different protein sources (plant and animal proteins), while maximizing maintaining the technological parameters of the production process. Products resulting from such activities can be of particular interest to people using different types of diets [4,5].

High-protein bars most commonly found on the store shelves contain a small range of proteins of both plant (primarily soy protein concentrates and isolates) and animal origin (especially whey protein concentrates and isolates). It was found that the addition of whey protein hydrolysates used in the application of high-protein bars has a positive effect on maintaining the soft structure of these products, but may affect the slightly bitter aftertaste [6]. Whey derivatives, such as concentrates or isolates, are abundant sources of proteins, in particular alpha-lactalbumin and beta-lactoglobulin. In the food industry, proteins of this type are widely used because of their high nutritional value, desirable sensory properties (milk flavor), and excellent functional properties [7]. For some time, however, there has been a sharp increase in interest in alternative protein sources (especially plant proteins) that could compete with the commonly used WPC protein in terms of physicochemical, textural or nutritional features [8].

Recently, plant proteins have been increasingly used as an economical and versatile alternative replacing animal source in human nutrition, as well as functional ingredients for product formulation. Animal protein presents growing costs and limited supply, which has been highly associated with climate change, freshwater depletion, biodiversity loss, and hazards for human health related to cardiovascular diseases etc. [9]. In addition, the use of plant proteins in food applications (including high-protein bars) may also increase the interest in these products among vegans, vegetarians, and people with an active lifestyle [10].

Examples of proteins that are not currently used on a large scale in Europe are e.g., sunflower, wheat, algae, hemp, and rice proteins. Sunflower seed is one of five major oil sources in the world. The defatted sunflower meals have relatively high content of protein, and have great economic value as a food additive. Due to low amounts of anti-nutritive compounds and no toxic substances found in these raw materials, they can be counted as a promising source for food proteins [11]. Sunflower protein may potentially be a functional protein due to relatively good solubility. It is often a by-product of oil extraction, which is usually denatured during processing and has reduced solubility and functionality. If protein fractions are isolated without being denatured, sunflower proteins may become soluble over a range of ionic strength and pH [10].

The wheat protein isolates are currently of special interest to processors and consumers due to low fat content and as a substitute for egg and dairy proteins. The major functional properties of wheat proteins are: hydration, foaming, improve sheeting properties of dough, light, and floppy texture as well as clean flavor [12]. Algae proteins have been valued around the world since the dawn of time for their nutritional value. Currently, there is an increasing interest in proteins of this origin also due to their functional and health-promoting properties (anti-inflammatory or enriching agents) [13]. Hemp proteins are characterized by a number of pro-health and pharmacological properties e.g., they affect the angiotensin I-converting enzyme (ACE) inhibition, renin inhibition, acetylcholinesterase (AChE) inhibition, metal-binding capacity, antioxidant activity, hypocholesterolemic effect, and serum glucose regulation and have significant amounts of arginine and glutamine [8].

Rice protein is gaining a lot of interest in the food industry due to its unique properties. Moreover, hydrolysis of protein with proteases could produce many potential peptide sequences providing numerous functional and antioxidative properties. It could also enhance the antioxidative properties of native protein by attacking the peptide bonds in the interior of polypeptide chains producing a range of polypeptides that differ in molecular weight or amino acid sequences [14].

Pumpkin proteins are an opportunity to use the large amounts of waste generated during the pumpkin seed oil pressing process. The oil cake obtained as a result of the process has a significant amount of amino acids. Moreover, the nutritional value of the protein preparations obtained from pumpkin is very high, and can be used to improve the nutritional value of food products [15,16].

Soy protein isolates (SPIs) are widely used in the food industry. Owing to the amphipathic (hydrophilic and hydrophobic) nature of soy proteins, SPI possess a favorable capacity to being adsorbed onto the oil-water or air-water interface and maintaining the structure of the corresponding system as stabilizers, i.e., SPI have good foaming and emulsifying abilities. Due to these properties and high nutritional value, low price, and availability, soy proteins are widely used [17].

Pea protein is one such protein that has garnered a great deal of interest based on its low allergenicity, high nutritional value, availability, and low cost. Similar to other plant proteins. However, challenges in utilizing pea protein as a food ingredient exist in terms of limitations in functionality, flavor, and color issues. Pea proteins contain high levels of lysine, but tend to be limiting in methionine and tryptophan. Accordingly, pea proteins are often consumed along with cereal grains, as they have a complementary essential amino acid profile in that cereal proteins are generally deficient in lysine but contain higher levels of sulfur amino acids (methionine, cysteine) [18].

Due to the characteristics of these proteins declared by the producers (nutritional value, content of saturated fatty acids, amino acid composition, fragmentation, etc.), they may be particularly interesting in research on WPC substitutes. 

Texture is one of the most important parameters that determines whether a product will be positively assessed by consumers and whether the customer will decide to buy a given product many times [19]. The degree of hardness in protein bars is directly correlated with the concentration of proteins used in the recipe. Too little protein may cause the formation of a liquid and ductile bar mass. On the other hand, overdosing the protein in the application will result in a loose and crumbling structure [20]. However, depending on the type of protein used, these parameters may differ from one other. 

The purpose of this article is to discuss the broadly understood physicochemical, textural, microstructural, nutritional, and sensory properties of proteins that may be alternatives to WPC proteins and their possible declarations in accordance with applicable European Union (EU) legislation. According to our current available knowledge, there is no research on ultrasonic viscosity and determining the color differences by Computer Vision System (CVS) for application of high-protein bars. Also, the use of alternative protein sources e.g., algae, pumpkin, hemp, sunflower or wheat in production of high-protein bars is limited. Therefore, the objective of this study was to evaluate the effect of different protein sources (algae, pumpkin, wheat, sunflower, rice, soy, hemp, pea, whey) on selected physicochemical, nutritional, and structural parameters of high-protein bars.

## 2. Materials and Methods 

### 2.1. Materials

Whey protein concentrate (WPC—80% proteins, 7.4% fat, 4.1% carbohydrates, granule size: <200 μm) was supplied by Polser Sp. z o. o. (Toruń, Poland), soy protein isolate (SPI—87% proteins, 3.1% fat, less than 1% carbohydrates, granule size: <200 μm) was purchased from Solae, pea protein isolate (PAP—82% proteins, 4% fat, 0.8% carbohydrates, granule size: <200 μm) was a product of Cosucra (Warcoing, Belgium), rice protein concentrate (RPC—80% proteins, 1% fat, 6% carbohydrates, granule size: <300 μm) was supplied by Barentz (Warsaw, Poland), wheat protein concentrate (WHP—77% proteins, 4% fat, 4% carbohydrates, granule size: <250 μm) was received from Cargill Polska (Warsaw, Poland), whole algal protein (ALP—60% proteins, 11% fat, 19% carbohydrates, granule size: <600 μm) was a product of TerraVia (San Francisco, CA, USA), sunflower protein (SUP—55% proteins, 2% fat, 9% carbohydrates, granule size: <200 μm), hemp protein (HMP—50% proteins, 10% fat, 5% carbohydrates, granule size: <500 μm) and pumpkin protein (PMP—60% proteins, 13% fat, 3% carbohydrates, granule size: <500 μm) were purchased from All Organic Trading (Wiggensbach, Germany), glucose syrup (Dextrose Equivalent “DE”40) was a product of Amylon (Havlíčkův Brod, Czech Republic), vegetable oil (rapeseed oil) was a product of ZT “Kruszwica” S.A. (Kruszwica, Poland), maltodextrin (Dextrose Equivalent “DE” 15) was purchased from Amylon (Havlíčkův Brod, Czech Republic), powdered barley malt extract was a product of Mountons Ingredients (European Brewery Convention “EBC” color: 5 to 12), soy lecithin (Identity Preserved “IP” 50) was supplied by Brenntag (Kędzierzyn-Koźle, Poland), natural vanilla aroma in powder was received from GBD Aromaty (Warsaw, Poland), chocolate was a product of Barry Callebaut (Łódź, Poland).

### 2.2. Preparation of High-Protein Bars

Protein concentrates (38.18%, *w*/*w*) with maltodextrin (5.45%, *w*/*w*) and aroma (0.91%, *w*/*w*) were placed in a bowl and mixed using the B10A industrial mixer (Technologies 4ALL Sp. z o. o. Sp. k.; Kępno, Poland) for 1 min at 190 rpm. Barley malt extract (3.64%, *w*/*w*) was dissolved in water (5.45%, *w*/*w*) in a separate laboratory vessel. In another vessel, soy lecithin (0.91%, *w*/*w*) and rapeseed oil (13.64%, *w*/*w*) were combined. Glucose syrup (31.82%, *w*/*w*) was heated to 80 °C and then poured into dry ingredients placed in the mixer bowl. The remaining ingredients prepared earlier were added simultaneously after pouring the syrup. The mass prepared in this way was mixed for 5 min at 365 rpm using an oar end. Finished processed high-protein bars mass were laid onto the conveyor belt using the CONBAR 600 (SOLLICH GmbH & Co. KG, Bad Salzuflen, Germany) and pulled out by forming rollers to a height of 15 mm. The height-adjusted bar mass was then cooled to a temperature of 10 °C for 15 min using the CONBAR 600 cooling tunnel. The chilled bar mass was subjected to longitudinal cutting using the CONBAR 600 longitudinal slitter. The longitudinally cut mass was finally cut transversely into individual bars (95 × 30 mm) using the CONBAR 600 transverse guillotine. The bars prepared in this way, which were to be coated with chocolate, were transferred to the coating machine, and they were covered with chocolate (22%, *w*/*w*) using the DK3520 (A.E. NIELSEN Maskinfabrik ApS., Farum, Denmark). Chocolate had a temper index (TI) oscillating in the range of 5.0–5.5. The parameter was measured using the Temper meter RET-250TMK (ELMI Automatic Systems, Warsaw, Poland). Chocolate coated bars were cooled to a temperature of 10 °C for 15 min using the DK3520 A.E. NIELSEN cooling tunnel. The final high-protein bars were packed in high barrier foil using a manual impulse sealer PFS 200 (NOVUMPACK, Kraków, Poland). The samples were stored at room temperature for 3 weeks in a plastic container. After the storage period, the samples were tested. The bar samples were unpacked from the foil 5 min before measurements. Cylindrical blocks of equal size (height: 15 mm, diameter: 12 mm) were punched out to analyze the texture of bars. To determine water activity and viscosity, the sample was prepared in the same way by weighing 6 g of sample for testing. Every high-protein bar sample was prepared in twenty repetitions. A total of 180 samples were tested considering all tests. The composition of tested high-protein bars without and with chocolate coating are presented in Table 1.

### 2.3. Texture Profile Analysis (TPA)

Texture measurements were carried out on TA-XT2i Texture Analyzer (Stable Micro Systems, Godalming, Surrey, UK) coupled with the Software Texture Expert. The test velocity was 1 mm/s. The high-protein bars were twice compressed by a 36 mm diameter probe (SMS P/36R) to achieve 70% deformation (interval between probe movements: 5 s). High-protein bar samples were evaluated for hardness, fracturability, adhesiveness, and cohesiveness. Analyses were carried out in five replications for each sample. The hardness value was determined as the peak force occurring during the first compression. Fracturability point was occurred where the plot has its first significant peak (where the force falls off) during the probe’s first compression of the product. Adhesiveness was calculated using the area over the negative stress–strain curve after the first compression, which represents the work per unit volume. Cohesiveness was defined as the ratio of the area under the second compression curve to the area under the first compression.

### 2.4. Cutting Test

Cutting strength of high-protein bars was measured using Texture Analyzer (TA-XT2i). The blade set with knife (HDP/BSK) comprising a Warner Bratzler blade (a reversible blade with knife edge) with a slotted blade insert and a blade holder was used for the experiment. In operation, the blade was firmly held employing blade holder, which was screwed directly to the texture analyzer. The slotted blade insert was placed directly onto the heavy-duty platform and acted as a guide for the blade whilst providing support for the product. High-protein bars were placed on the metal plate. Then the blade was lowered at a speed of 2 mm/s. The cutting curve was obtained by recording the maximum force the blade needs to cut the sample completely. Five repetitions were applied for each formulation. The results were based on the maximum peak (maximum force) resulting from the shear stress.

### 2.5. Water Activity

Water activity (a_w_) was measured using the AWMD-10 water activity meter (NAGY, Gäufelden, Germany) with the accuracy of ±0.001 of a_w_ unit. Before measurement, the apparatus was calibrated with the dedicated humidity standard (95% HR). Measurements were performed at the temperature of 25 °C, in five repetitions. For each sample, two outliers were classified as defective and were excluded from further analysis.

### 2.6. Computer Vision System (CVS) and Determining Color Differences

Computer Vision System (CVS) was applied according to Tomasevic et al. [21] with the use of Sony Alpha DSLR-A200 digital camera (10.2 Megapixel CCD sensor, SONY, Tokio, Japan). The color was expressed in terms of the International Commission on Illumination (CIELAB) color space with the coordinates being L* (0–100, estimation of lightness), a* (red-green) and b* (yellow-blue) [22]. The noted differences could be described as “marked changes” according to the NBS (National Bureau of Standard) reference scale, which implies that such changes are perceptible to the human eye.

The total color difference was calculated using the formula:$$\Delta E=\sqrt{{\left(a_{1}-a_{2}\right)}^{2}+{\left(b_{1}-b_{2}\right)}^{2}+{\left(L_{1}-L_{2}\right)}^{2}}$$

Moreover, the ΔE* values were converted into National Bureau of Standards (NBS) units by the equation [23]:NBS units=ΔE×0.92

### 2.7. Ultrasonic Viscosity

The dynamic viscosity of high-protein bars was measured using an ultrasonic viscometer Unipan type 505 (UNIPAN, Warsaw, Poland). Measurements of the viscosity were performed at 25 °C. Prior to each measurement, the ultrasound signal level was checked. The measuring probe was immersed completely in the high-protein bar. The results were read in mPas·g/cm^3^. All measurements were performed in three repetitions.

Viscosity tests using ultrasounds rely on the use of a magnetostrictive probe, which produces free vibration [24]. An alternating electric current generates an alternating magnetic field that causes the phenomenon of magnetostriction, i.e., the deformation of ferromagnetic materials. Induced ultrasound waves are damped by the tested material. Ultrasound viscometers display the result as the product of viscosity and density [25]. Ultrasonic viscometer viscosity measurements are performed at high frequency, and for this reason, it is not easy to compare the obtained results with those obtained using other viscometers. In addition, ultrasonic viscometers are used for continuous measurements of viscosity under conditions where measurements can be difficult and it is not possible to use devices such as rotational viscometers [26].

### 2.8. Nutritional Value

Nutritional value was calculated based on raw material specifications obtained from suppliers of each of the ingredients, which was introduced into the program. Then the recipe was entered into the X-mart (X-mart Group Sp. z o. o., Lublin, Poland) software and the nutritional value of the finished product was calculated per 100 g.

### 2.9. Sensory Evaluation

A panel of 15 trained consumers was recruited from EUROHANSA Sp. z o. o. The criteria for selection were that the panelists should be between 18 and 60 years old and regular consumers of high-protein bars and not allergic to any raw material used. Panelists were instructed to evaluate the sensorial attributes; color, aroma, consistency and taste. A 5-point hedonic scale (1 = extremely dislike, 5 = extremely like) with significance factors (0.2—color, 0.2—aroma, 0.25—consistency and 0.35—taste) was used [27,28].

### 2.10. Heavy Metals Analysis

The obtained samples were grounded and about 0.5 g of the sample were weighed from the homogeneous mass on an analytical balance with an accuracy of 0.0001 g. After the tubes were closed, they were transferred to the mineralizer rotor. The mineralization was carried out in a CEM Mars Xpress microwave oven at the temperature of 210 °C and pressure of about 7 atm. The obtained clear mineralizates were quantitatively transferred to 50 cm^3^ volumetric flasks and diluted with demineralized water (conductivity 0.055 µS/cm) to the mark. The obtained solutions were analyzed on an inductively coupled plasma mass spectrometer (ICP Mass Spectrometer Varian MS-820, Santa Clara, CA, USA). The gas used to generate the plasma was argon from Messer with a purity of 99.999%. No reaction chamber (CRI) was used in the analysis. The following camera settings were used: Plasma Flow—16 dm^3^/min., Flow Nebulizer—0.98 dm^3^/min., RF Power—1.38 kW, Sampling Depth—6.5 mm. The following isotopes of the analyzed elements were used: ^114^Cd, ^206^Pb, ^207^Pb, ^208^Pb.

The determination was made using the standard curve method. Ultra Scientific standards with a purity of 99.999% were used for the analysis. The results obtained are expressed in mg/kg fresh weight. Test quality control during the analysis was applied by measuring blank, duplicate and certified reference material “NIST-1577c Bovine Liver”.

### 2.11. Amino Acids Determination

The sample (approx. 70 mg of pure protein) was hydrolyzed with 6 N HCl at 110 °C for 20 h. After cooling, the solution with the sample was filtered through a G-5 funnel. 4 mL of the hydrolysate was evaporated in a vacuum evaporator. The dry residue from the vacuum flask was dissolved in 5 mL of citrate buffer pH 2.2. The prepared sample was dispensed onto the amino acid analyzer column [29].

Separation of sulfur amino acids was performed as follows: cysteine was oxidized to cysteic acid, and methionine to methionine sulfone using performic acid. The mixture was then flooded with 1 mL of 40% HBr and concentrated in a vacuum evaporator. It was then quenched with 6 N HCl and hydrolyzed at 110 °C for 20 h. Further procedure was the same as for protein amino acids [30].

To determine tryptophan, the sample was subjected to alkaline hydrolysis. The sample weight containing approx. 75 mg of protein was hydrolyzed in Ba(OH)_2_ solution at 110 °C for 20 h. The sample was then acidified with 6 N HCl and a Na_2_SO_4_ solution was added. The contents were transferred to centrifuge flasks and centrifuged for 15 min at 3000× *g*. The supernatant, after filtering through a syringe filter, was dosed into the amino acid analyzer.

Amino acids were determined with the AAA 400 amino acid analyzer from Ingos (Prague, Czech Republic). Amino acids were separated through ion exchange chromatography. The column with dimensions of 0.37 × 45 cm was filled with an ion exchanger in the form of a resin. LG ANB ostium was used for the hydrolysates. It is a strong cation exchanger with an average grain size of approx. 12 µm in the form of Na (column temperatures 60 °C and 74 °C). The apparatus detects the amino acids by ninhydrin derivative (this is the detection reagent). The identification of the amino acids was performed by a photometric detector at a wavelength of 570 nm for all amino acids, while for proline—440 nm. Four buffers were used for separation: 1—pH 2.6, 2—pH 3.0, 3—pH 4.25, 4—pH 7.9. After the amino acid separation, the column was regenerated using 0.2 N NaOH.

### 2.12. Scanning Electron Microscopy (SEM)

The samples were placed in a 4% aqueous glutaraldehyde solution at room temperature for 2 h and then transferred to a refrigerator at ca. 4 °C for 6 h. After this time, the samples were placed in Sörensen’s phosphate buffer pH 7.0 and left overnight. After removing from the buffer and washing twice in distilled water, the samples were dehydrated in an acetone series. Concentrations of acetone solutions (p.a.), to which the samples were successively transferred, were: 15, 30, 50, 70, 90, 100%. The samples were kept for 30 min in each of the solutions. At the end of the dehydration process, the samples were placed in anhydrous acetone dried on silica gel for 30 min. The last stage was carried out twice. For final removal of residual water, the material was subjected to critical point drying with carbon dioxide in an Emitech K-850 dryer (Ashford, UK). Microscope tables were prepared with a carbon substrate placed on them and dried samples of bars were attached to it. The prepared gold was sputtered with an Emitech K-550X sputter (Ashford, UK). After the preparation was completed, the obtained material was placed in a Tescan Vega LMU (Brno, Czech Republic) scanning electron microscope and examined under high vacuum.

### 2.13. Statistical Analysis

Statistical analysis was carried out with a help of the STATISTICA 13.3 PL software (Stat Soft Polska Sp. z o. o., Kraków, Poland). A one-way ANOVA analysis was performed, and significant differences between samples were determined applying the Tukey *post hoc* test at *p* < 0.05.

## 3. Results and Discussion

### 3.1. Texture Profile Analysis (TPA), Cutting Test and Scanning Electron Microscopy (SEM)

The influence of different proteins on hardness, fracturability, adhesiveness, and cohesiveness of the obtained processed high-protein bars with or without chocolate coating is presented in Table 2a,b. Significant differences (*p* < 0.05) were observed. The bar made of sea algae proteins (ALP) had the highest hardness (276.43 N with chocolate and 288.50 N without chocolate coating) in both cases, while the lowest hardness characterized bars made from pea protein (PAP) in the sample without chocolate (13.62 N) and made from rice protein (RPC) in the sample with chocolate coating (20.95 N).

Sensory hardness can be defined as the force necessary to compress a high-protein bar with the molars. Fracturability is the tendency of a material to fracture, crumble, crack, shatter or fail upon the application of a relatively small amount of force or impact. Adhesiveness is the work/force necessary to overcome the attractive forces between the surface of a product and the surface of a material (the probe), with which the product comes in contact. Cohesiveness is the tendency of a product to cohere or stick together [31]. Generally, hardness of high-protein bars is quite high and increases with the addition of protein [32]. The developed high-protein bars are characterized by large variety of parameters. Considering the research of Banach et al. [33], a certain regularity can be noted for bars made of whey proteins. Relatively low value of the hardness parameter translates into high values of the adhesiveness and cohesiveness, and simultaneously low levels of fracturability. A completely different situation can be seen for algae protein. Despite the high hardness of the bar made of this type of protein, high values of the adhesiveness and cohesiveness parameters as well as very low fracturability values were observed in the chocolate-coated bar. The results obtained from the research on a bar made of sunflower proteins (SUP) also deserve attention due to relatively high level of all TPA parameters, in particular in the chocolate-coated sample. The intermolecular attraction, by which the elements of a body or mass of material are held together, determine its cohesiveness [34]. Banach et al. [35] found a connection between the hardness of bars and the size of the protein particles used in the recipe for making those bars. Based on the analysis of results in the Table 2a,b and pictures of the microstructure of bars (Figure 1), it can be assumed that proteins with large particle sizes used in the production of high-protein bars caused a significant increase in the hardness of the final product. According to this lead, fine-grained proteins have much lower tendency to form hard structures during the storage process and allow for the creation of a delicate and soft product structure, which is confirmed by Cho [36]. It was observed that the chocolate-coated bars showed, in most cases, higher hardness than the uncoated samples, but also had increased other TPA parameters. This is most likely related to the greater restriction of air access to these products, which slows down the drying processes of the product (as evidenced by higher adhesiveness results and water activity parameters presented in Figure 2a,b). 

Cutting test indicates the firmness/hardness of a product. If one considers that if the top front teeth were pulled from a curve-shape into a straight line, they would represent a ‘knife edge’. Using a knife blade gives a close representation of the biting or cutting action [31].

With reference to Table 3, the bar made of algae proteins (ALP), both without and with chocolate coating, showed the highest cutting resistance (166.82 N without chocolate and 235.45 N with chocolate). Whereas, the most susceptible to this effect turned out to be high-protein bars made of proteins: wheat (WHP: 8.43 N without chocolate and 10.54 N with chocolate), pumpkin (PMP: 7.69 N without chocolate and 22.7 N with chocolate), hemp (HEP: 10.58 N without chocolate and 15.59 N with chocolate) and sunflower (SUP: 14.38 N without chocolate and 22.35 N with chocolate coating). The obtained results correlate with the hardness analysis and confirm the relationship between the hardness of bars and the force needed to cut them. In general, it can be observed that the chocolate-coated bars had higher resistance to the cutting force, except from WPC and PAP. The reason for this phenomenon may result from slight fluctuations in parameters in the degree of chocolate tempering, which slightly change during the chocolate tempering process. Chocolate with a temper index (TI) degree of 5.0–5.5 is characterized by high hardness causing a characteristic crackle when breaking, which is a feature desired by consumers in this type of products. This is also confirmed by the sensory analysis performed. The deviations may be related to the thickness of the chocolate layer in different places of the bars because in production conditions the products are coated with a stream of chocolate, too high layer of chocolate is blown off through a blast of compressed air, which creates a wave on the product characteristic of chocolate-coated products available on the store shelves. A properly tempered chocolate exhibits high gloss, appropriate melting temperature, and fat-bloom stability with desired characteristic crunchiness and hardness during eating depending on the amount of chocolate on the final product [37]. In addition, the flow behavior of tempered chocolate has implications for the processing of chocolate after tempering. Factors such as conching temperature, particle size distribution, fat content, type of emulsifiers, and tempering conditions determine efficiency of mixing, pumping, and transportation of final products during processing [38].

Based on the electron microscope photos presented in Table 3 and the studies by Labuza and Hyman [39], it can be assumed that large discrepancy in the results of TPA tests may be due to factors related to the structural features, density of protein molecules and their porosity. In addition, Hogan et al. [6,40] proved that the rate of moisture migration in multi-domain foods is slower in foods with smaller pore size, presumably due to more tortuous pathways for moisture diffusion. Air occluded within powder particles may be observed as the proportion of powder volume not subjected to moisture-induced change, and thus may be beneficial to structural stability. A fairly large number of similarities were found between the microstructure of SUP, PAP, WPC and RPC proteins. Pictures of these proteins have the significant number of depressions and embossing. Taking into account the parameters of the texture analysis, such a structure probably has a significant impact on decreasing the hardness and cutting force parameters. Bars with SUP revealing much higher hardness, were the exception. This may be related to the formation of large clusters (agglomerates) of proteins, resulting in the formation of a compact and hard structure, which translates directly into high parameters of hardness, fracturability, and adhesiveness. It was also observed that all the above bars were characterized by quite high susceptibility to the action of shear force. It is worth mentioning that bars made of WHP and PMP proteins, having a wavy structure, without a large number of cavities and air pores, were also characterized by relatively low hardness parameters and high susceptibility to cutting. All bars made of the proteins mentioned above showed similar, technologically acceptable, water activity (a_w_ < 0.735). The bar made of ALP proteins had a very diverse structure. Numerous air pores and unevenly distributed agglomerates of protein particles in the form of bushy protrusions were observed on its surface. Interestingly, the protein protrusions were also interrupted by wavy, relatively smooth protein structures. Referring to the work of Bleakley and Hayes [41], the formation of characteristic agglomerates in the case of algae protein may be related to the presence of lectins in this type of protein. Lectins are glycoproteins known for their aggregation and high specificity binding with carbohydrates without initiating a modification through associated enzymatic activity. Completely different microstructure of ALP bars is probably the reason for the highest hardness. The use of proteins of various botanical origin in the study has a significant impact on the swelling method, reorganization of molecular structures or aggregation of proteins in the product. However, the control of micro- and macrostructures is still very difficult due to the poor knowledge of this issue [42].

### 3.2. Water Activity

Measurement of water activity of high-protein bars are presented on Figure 2a,b. The water activity of high-protein bars changes during the storage process [43]. Therefore, the bars were stored for 3 weeks in a sealed plastic container (metallized barrier film) at 20 °C. The optimal storage time was adopted due to the results of studies by Banach et al. [35], which showed that a_w_ increase after this period was not significant. Water activity of the tested samples was shown in Figure 2a,b. The highest a_w_ value characterized bar with chocolate coating made of sunflower protein (SUP)—0.735 and the lowest (ALP)—0.530 algae protein bar without chocolate coating. Bars made of proteins: sea algae (ALP), pea (PAP) and soybean (SPI) had a_w_ lower than 0.65, which guarantees the stability of samples during storage (in room temperature) and inhibition of microbial growth [44]. Other bars, made of hemp (HMP), rice (RPC) and sunflower (SUP), had water activity above 0.65. Therefore, it can be suggested that they should be stored in conditions of lower temperature. Increased water activity may indicate movement of water molecules from the intermediate phase, where they act as a plasticizer, to the bulk phase [32]. Proteins included in bars that exceeded 0.65 water activity values, had a fairly high degree of fragmentation, and originated from various plant species, which could also have a significant impact on this parameter.

According to the current knowledge, changes in the structure or organization of proteins in a product may be associated with the formation of disulfide bonds when there are no water molecules associated with the local protein domain. It may also be one of the mechanisms explaining the hardening of high-protein products over time, in particular, if whey proteins have been added to the product [45,46]. High water activity parameters for WHP without chocolate coating may be related to the high water absorption of gluten. On the other hand, the differences in the case of the sample with chocolate coating may result from the restriction of air access to the interior of the product due to the coating with a layer of chocolate. High water activity value of bars made of SUP protein may be related to the high capacity of this type of protein to absorb water and fat, as evidenced by the research of Ren et al. [47]. According to the obtained results, it can be suspected that the addition of other types of proteins, such as ALP, PAP, WHP, or SPI to food products, may have a positive effect on reducing the water activity and modifying the textural parameters of the final product.

### 3.3. Color Differences Measured with Computer Vision System (CVS)

Typically, the main aspect that reduces the quality of high-protein bars is their hardening over time, but color can also be an important quality indicator for consumers of this type of product. The colors of bars made of various types of proteins after three weeks of storage are presented in Table 4a. The most frequently used methods of color assessment in the analysis of this parameter in food products are colorimetric methods. In turn, the colors generated using CVS closely resemble the real color of the samples being assessed. Moreover, the color is more intense (the colors are more saturated) than for standard colorimetric methods [48]. In this study, the CVS method was used due to the studies by Tomasevic et al. [49], which proved that the CVS method gives much better results in assessing the color of food products.

According to Inami et al. [23], all the color differences expressed by values larger than 6 are considerable. Discrepancies between the >6 prove the high impact of the protein used on the color difference of high-protein bars. Proteins from various sources were characterized by different colors compared to the blank sample made of WPC. The SPI protein was characterized by the highest brightness, which may indicate a slower ability to bind fat than other tested proteins. The fat on the surface of bars probably causes the increased ability to reflect light. The remaining differences in parameters a and b are probably directly related to the origin of the component (plant and animal proteins). Considering the Hasan [50] study, values of the L* parameters were similar for bars made of whey proteins after about 6 weeks of storage. Minor differences in the a* and b* parameters may be related to the analysis of parameters after a longer storage period. On the other hand, comparing the research results with the work of McMahon et al. [51], it can be assumed that bars made with high-fructose and glucose syrups are much more susceptible to darkening processes than in the case of other types of syrups. In this study, only glucose syrup was used in all trials. Therefore, it is probably required to conduct further tests and investigate whether the type of syrup will cause large changes in the L* a* b* parameters for the developed high-protein bars. The color of bars with chocolate coating was not examined, but it can be assumed that they would have brighter colors due to better protection against light and air access, and thus slowing down the Maillard reaction.

### 3.4. Ultrasonic Viscosity

The obtained ultrasonic viscosity is presented in Figure 3a,b. The ultrasonic viscometer gives the results of measurements in units of dynamic viscosity multiplied by density. The highest viscosity values were recorded for bars made of WHP and HEP proteins. However, the lowest—for ALP, WPC, PAP, and SUP. Low values of the viscosity parameter, in particular for ALP, correlate with high results of the hardness parameters and cutting force. It is also worth paying attention to the microstructure of these proteins, in which numerous clusters of wide pores (probably fat-air), a tightly compact and irregular structure can be seen (Figure 1). 

On the other hand, high-protein bars with the highest lightness values had much lower values of parameters related to hardness and cutting force. Their microstructure is free from numerous pores and the surface is much more even compared to other tested samples. Considering the obtained results and comparing them with the research by Tomczyńska-Mleko and Ozimek [52], it can be assumed that the obtained results could be influenced by factors such as degree of aeration in the bar mass and the consistency and protein concentration the product was made of. Ultrasonic viscosity tests are a rare/novel method for the analysis of food products, therefore, according to current knowledge, it is difficult to find publications to compare the obtained results.

### 3.5. Heavy Metals Analysis

The obtained results of testing the content of heavy metals in the developed high-protein bars are presented in Figure 4. They do not exceed the current permissible concentrations for this type of products according to Commission Regulation (EC) No 1881/2006 of 19 December 2006 [53].

However, a much higher level of cadmium and lead is very visible for the RPC protein. Regarding the research of Huang et al. and Kaneta et al. [54,55], it can be assumed that the increased level of these elements for bars made of RPC may result from the ability of rice, one of the most commonly cultivated plants on earth, to absorb significant amounts of heavy metals from its cultivation sites. Therefore, products made of this type of grain may be characterized by an increased content of not only cadmium and lead, but also arsenic and other heavy metals.

### 3.6. Amino Acids and Nutritional Value

Often, for people consuming the high-protein products (e.g., athletes, sick people or convalescents), nutritional and pro-health values are important. The obtained results of the content of amino acids in the tested bars presented in Figure 5 show significant differences between the content of individual amino acids in different types of proteins. The largest deviation can be seen in the proline content for bars made of WHP protein.

Based on the work of Kowieska et al. [56], it can be assumed that such a high content of endogenous amino acid, i.e., proline, may be caused by a naturally high content of proline in wheat grain. Proline is an important amino acid for physically active people and athletes, as it participates in the formation of secondary structures in collagen. These structures are stabilized by enzymatic hydroxylation (proline hydroxylase) or a substituent having electron withdrawing ability, e.g., fluorine, which significantly increases the stability of the collagen. Deficiencies of proline, vitamin C (being a cofactor of proline hydroxylase), and disorders of enzyme production, can lead to the scurvy [57].

Exogenous amino acids are essential and human body cannot synthesize them from scratch at a rate commensurate with their needs. Therefore, it is necessary to provide them to the body in a properly balanced diet. Out of a total 21 amino acids, nine are considered essential, including phenylalanine, valine, threonine, tryptophan, methionine, leucine, isoleucine, lysine, and histidine. Proteins found in animal sources, such as meat, poultry, fish, eggs, milk, cheese, and yogurt, provide all nine indispensable amino acids and hence they are referred to as “complete proteins.” Proteins found in plants, legumes, grains, nuts, seeds, and vegetables tend to be deficient in one or more of the indispensable amino acids and are called “incomplete proteins” [58].

Despite the passage of time and thermal processing during production, the obtained high-protein bars are characterized by a high content of the entire spectrum of amino acids.

Considering the obtained results, it can be assumed that, despite the high content of essential amino acids in the WPC reference sample, alternative sources of plant-derived proteins may be an attractive proposition for people who do not want or cannot consume proteins of animal origin. It is worth paying attention to the high content of arginine in the ALP. As evidenced by the research of Stróżyk et al. [59], arginine is an essential amino acid in the case of increased physical exertion. It is a precursor of nitric oxide, which relaxes the smooth muscles of blood vessel walls, thus improving the blood oxygenation and replenishment of skeletal muscles. It was observed that bars made of SPI, RPC, and PAP proteins showed significant alignment of the entire spectrum of amino acids. Among which, essential amino acids, which have been determined in significant amounts, as for proteins of plant origin, deserve special attention. The high content of essential amino acids in soybean and rice protein isolates was also confirmed by Kalman [60]. Therefore, it can be assumed that these proteins may become more and more popular, especially among vegans and vegetarians and allow the essential amino acids in such diets to be satisfied.

Results obtained from the calculated nutritional value presented in Figure 6a,b and Figure 7a,b indicate slight differences in the protein and fat content as well as nutritional values in individual protein preparations used in the production of high-protein bars. However, these differences are not significant (*p* > 0.05). It is also worth paying attention to the fiber content for bars made of HEP and SUP, the content of which is significantly higher than that of other samples. In addition, this amount of fiber allows the nutrition claim “high fiber content” to be used on the packaging of these products in accordance with Regulation (EC) No 1924/2006 Of The European Parliament And Of The Council of 20 December 2006 [61] on nutrition and health claims made on foods. The consumption of fiber is an important aspect in the diet of every human being due to many positive functions of fiber. Local reactions are related to their presence in the gastrointestinal tract and systemic reactions with an effect on metabolism. Viscosity, the ability to ferment, binding water, binding bile acids, reacting with metal ions, and increased stool weight are just a few of better understood effects of dietary fiber consumption Li and Komarek [62]. Based on Figure 6a,b, it can be concluded that the developed high-protein bars had a low content of saturated fatty acids. This is an important feature from the point of view of athletes and physically active people because saturated and trans fats may also make the lining of blood vessels (the endothelium) less flexible. In addition, trans fats may depress the “good” blood cholesterol (HDL cholesterol) when eaten in large quantities [63].

### 3.7. Sensory Evaluation

Results of the sensory evaluation of the tested high-protein bars are presented in Figure 8a,b. The highest scores during the analysis were obtained for bars made of WPC and PAP proteins. The evaluators appreciated the external appearance, color and taste sensations the most in the highest rated bars. High ratings for these types of proteins are associated with a pleasant consistency, taste and color, according to the judges. The worst rated bar (ALP) had too high hardness, an unpleasant aftertaste and a green-yellow color, which gave it the lowest scores. Negative assessments of the taste of the ALP bar may also be caused by too much dosing of this type of protein for a given application. Based on research by Hall et al. [64], in which the addition of algae protein in bread was 4%, it can be suspected that low taste ratings of bars made of algae proteins were probably caused by too high percentage in the final product (30%, *w*/*w*). In contrast, the research of Prabhasankar et al. [65] prove that the addition of algae protein in the amount >10%, *w/w* in pasta causes acceptable sensory features for those taking part in the sensory evaluation. Considering the above results, it can be concluded that the chocolate coating of high-protein products significantly increases their palatability. Ratings for individual chocolate bars were clearly higher than their non-chocolate counterparts. The remaining high-protein bars were rated at an average level with a tendency to be more positive. Referring to the research by Usha et al. [66], in which the effect of adding the pumpkin flour to baby food after weaning was examined, the ratings of products, in which addition was 20–30%, *w/w* pumpkin flour, were on the average level (on a nine-point scale) with a tendency to higher one, in particular if dosing at the level of 10%, *w*/*w*. It is worth mentioning that the worst rated parameter by the team was the taste of dishes, which can be correlated with the results obtained from the study of high-protein bars.

## 4. Conclusions

Based on the experiment, it can be concluded that changes in textural parameters, nutritional values or physicochemical parameters significantly depend on the type of protein used. Differences in the parameters of texture profile analysis (TPA) and cutting forces show significant differences, of which bars made of RPC, HEP, PAP, and PMP were characterized by the lowest results during the TPA test and ultrasonic viscosity. Differences in the microstructural structure of the tested bars significantly translated into the physical, chemical and textural features presented by proteins. It can be suspected that the microstructure of proteins has a significant effect on the water activity of high-protein bars. In particular, ALP proteins showed the lowest results of this parameter (a_w_ < 0.55), which ensures the microbiological stability of the final product, even during long storage periods. The obtained results of the amino acid content give the possibility that the SPI, RPC, and PAP proteins can compete with the WPC protein in terms of essential amino acid content. Bars made of RPC proteins had an increased content of heavy metals but these values did not exceed the acceptable EU standards. In terms of the nutritional value studied, all types of protein deserve attention due to the low content of saturated fat. Bars made of SUP and HEP proteins, apart from high protein content, also allowed for the declaration of high fiber content (fiber content above 6 g/100 g). Based on sensory analysis and the color assessment using the CVS method of bars without chocolate coating, it can be concluded that the color plays an important role for the consumer. Sensory analysis showed that the coating of high-protein bars with chocolate increases scores of the tested products, masking specific smells, color to a large extent, the aftertaste of some types of proteins, thus contributing to an increase in the overall sensory assessment. Therefore, it can be clearly defined that high-protein bars should be covered with chocolate. The conducted research shows that proteins of plant origin can be successfully used in the food industry as an alternative to WPC proteins, however it is not possible to clearly indicate which type of protein is the best option. Nevertheless, the high content of exogenous amino acids or the technological utility resulting from the texture parameters of bars made of RPC, SPI and PAP proteins speak for these sources. More research is needed on storage trials, microbiological tests, and examining the effect of changing other components in the recipe that may affect the parameters important for the food industry.

## Figures and Tables

**Figure 1 foods-09-01467-f001:**
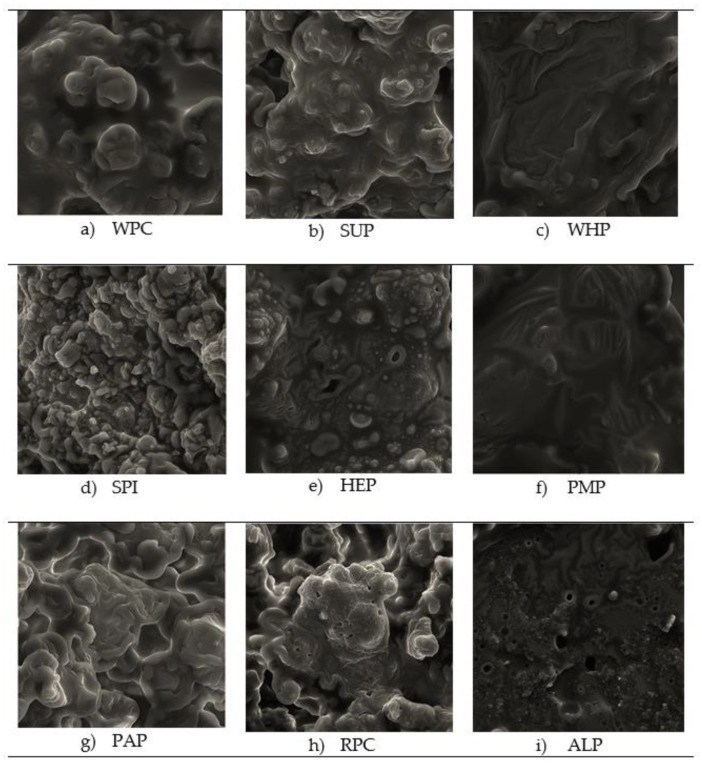
(**a**–**i**) Microstructure of high-protein bars from the scanning electron microscope (SEM HV: 30 kV, View field: 271 μm, SEM MAG: 800×).

**Figure 2 foods-09-01467-f002:**
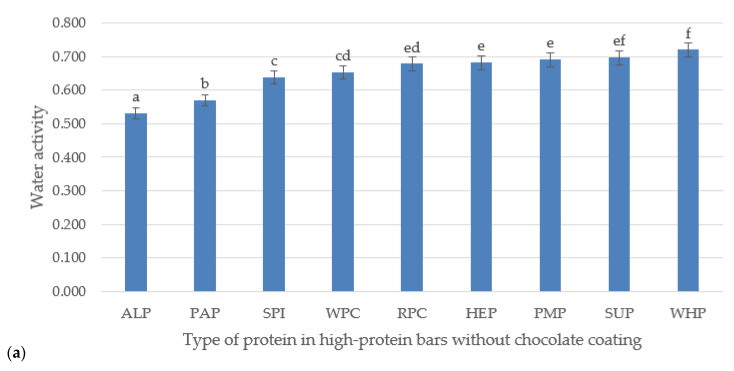
(**a**) Effect of different protein application on water activity of high-protein bars without chocolate coating; (**b**) With chocolate coating. Letters (a–f) indicate significant differences at *p* < 0.05 (Tukey’s HSD test).

**Figure 3 foods-09-01467-f003:**
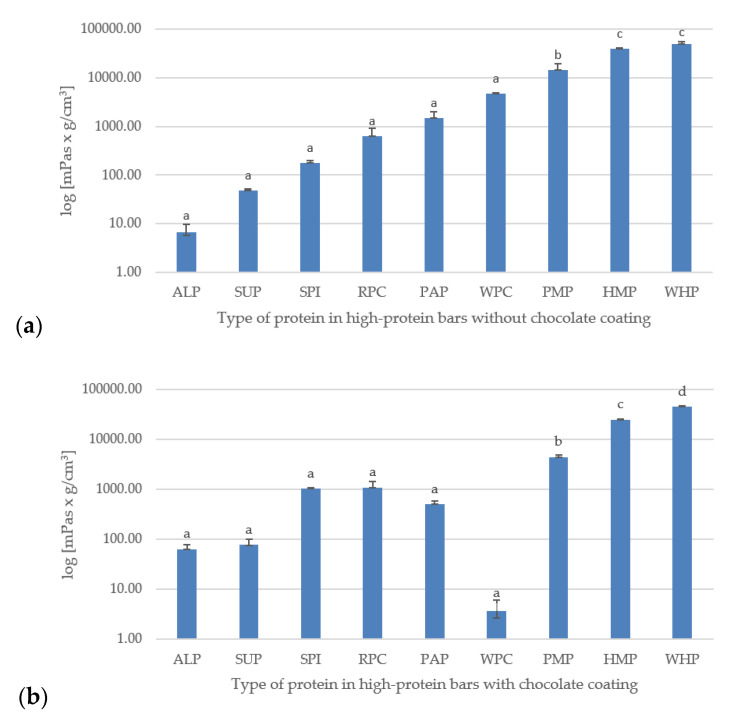
(**a**) Ultrasonic viscosity measurement results (mPas g/cm^3^) of tested high-protein bars without chocolate coating; (**b**) With chocolate coating. Letters (a–d) indicate significant differences at *p* < 0.05 (Tukey’s HSD test).

**Figure 4 foods-09-01467-f004:**
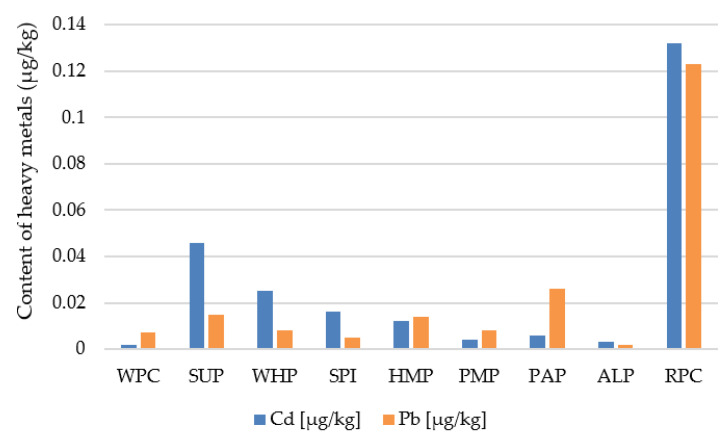
Content of heavy metals (cadmium and lead) in the high-protein bars.

**Figure 5 foods-09-01467-f005:**
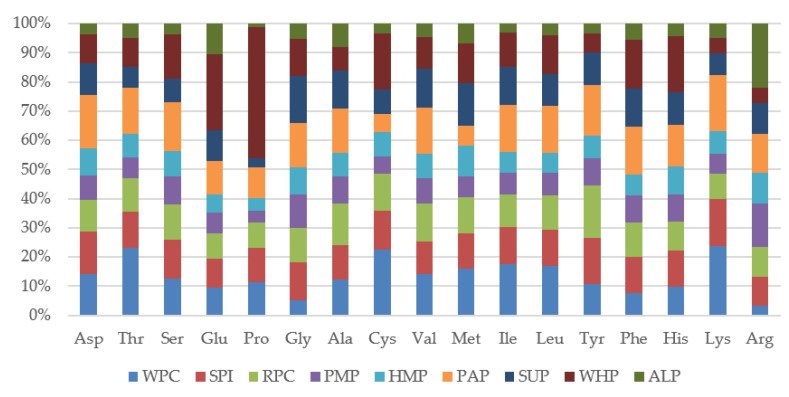
Percentage of individual amino acids in the high-protein bars.

**Figure 6 foods-09-01467-f006:**
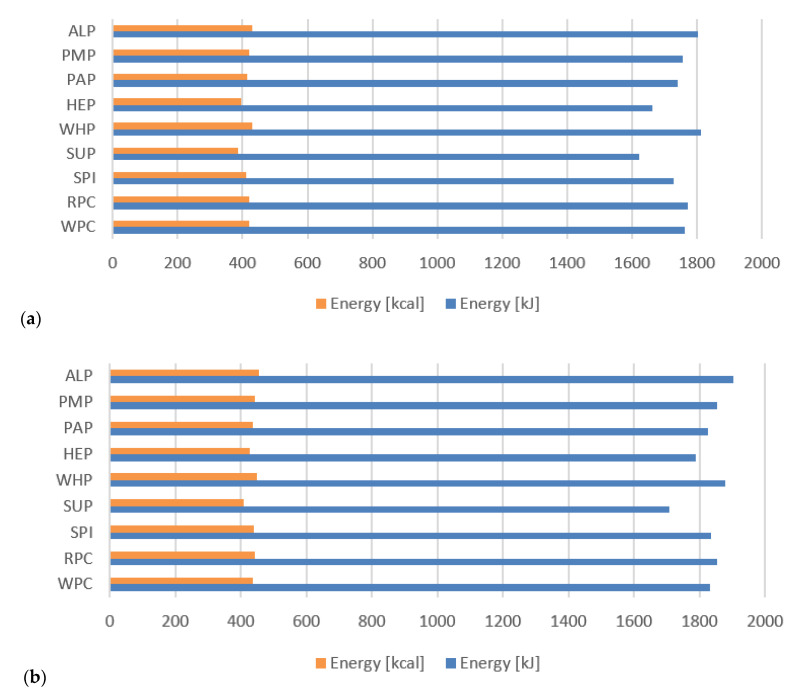
(**a**) Energy value of high-protein bars without chocolate coating; (**b**) With chocolate coating.

**Figure 7 foods-09-01467-f007:**
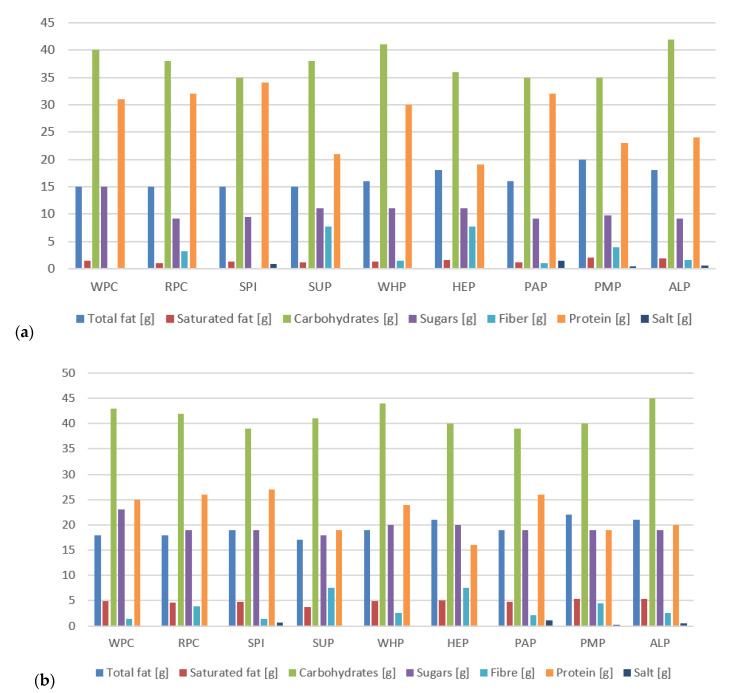
(**a**) Nutritional value of high-protein bars without chocolate coating; (**b**) With chocolate coating.

**Figure 8 foods-09-01467-f008:**
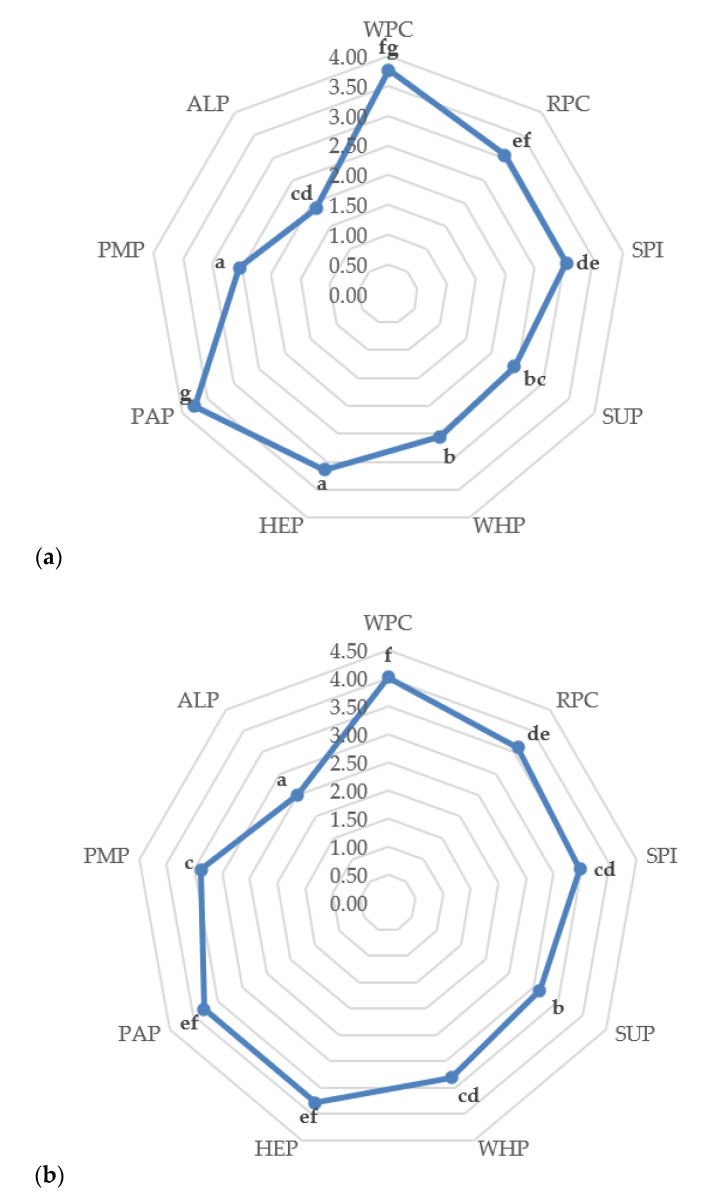
(**a**) Sensory evaluation of high-protein bars without chocolate coating; (**b**) With chocolate coating. ^a–g^ Means in the same column with different superscripts are significantly different (*p* < 0.05, Tukey’s HSD test).

**Table 1 foods-09-01467-t001:** Composition of high-protein bars.

**Composition of High-Protein Bars without Chocolate Coating**	
**Ingredient**	**Percentage Content in Final Product (% *w*/*w*)**
Protein ingredient (WPC, SPI, PAP, RPC, WHP, ALP, SUP, HMP or PMP)	38.18
Glucose syrup	31.82
Rapeseed oil	13.64
Maltodextrin	5.45
Water	5.45
Barley malt extract	3.64
Soy lecithin	0.91
Vanilla flavor (aroma)	0.91
**Composition of High-Protein Bars with Chocolate Coating**	
**Ingredient**	**Percentage Content in Final Product (% *w*/*w*)**
Protein ingredient (WPC, SPI, PAP, RPC, WHP, ALP, SUP, HMP or PMP)	30.2
Glucose syrup	25.0
Rapeseed oil	10.8
Maltodextrin	4.3
Water	4.3
Barley malt extract	2.9
Soy lecithin	0.7
Vanilla flavor (aroma)	0.7
Chocolate	21.1

**Table 2 foods-09-01467-t002:** Effect of different protein application on high-protein bars texture attributes. (**a**) Data are presented as means ± SD (standard deviation). ^a–h^ Means in the same column with different superscripts are significantly different (*p* < 0.05, Tukey’s honest significant difference “HSD” test). (**b**) Data are presented as means ± SD (standard deviation). ^a–i^ Means in the same column with different superscripts are significantly different (*p* < 0.05, Tukey’s HSD test).

**(a)**				
**Type of Protein Used in Bars with Chocolate Coating**	**Texture Attributes**			
	**Hardness (N)**	**Fracturability (N)**	**Adhesiveness (J)**	**Cohesiveness**
WPC	54.66 ^e^ ± 0.303	0.06 ^a^ ± 0.005	382.87 ^g^ ± 4.977	0.14 ^f^ ± 0.001
RPC	20.95 ^a^ ± 0.164	0.06 ^a^ ± 0.004	57.54 ^c^ ± 2.588	0.07 ^c^ ± 0.001
SPI	25.25 ^c^ ± 0.358	20.85 ^b^ ± 0.152	66.85 ^c^ ± 1.773	0.04 ^a^ ± 0.004
SUP	136.61 ^g^ ± 0.406	115.56 ^f^ ± 0.255	225.61 ^f^ ± 4.861	0.10 ^de^ ± 0.001
WHP	88.33 ^f^ ± 0.092	55.38^e^ ± 0.288	123.42 ^d^ ± 3.724	0.11 ^e^ ± 0.004
HEP	27.74 ^d^ ± 0.152	0.03 ^a^ ± 0.005	27.15 ^b^ ± 2.528	0.13 ^f^ ± 0.003
PAP	27.44 ^d^ ± 0.302	25.86 ^d^ ± 0.461	145.78 ^e^ ± 4.853	0.06 ^b^ ± 0.002
PMP	23.52 ^b^ ± 0.338	23.27 ^c^ ± 0.215	7.34 ^a^ ± 0.392	0.09 ^d^ ± 0.004
ALP	276.43 ^h^ ± 0.286	0.13 ^a^ ± 0.012	129.67 ^d^ ± 0.577	0.19 ^g^ ± 0.008
**(b)**				
**Type of Protein Used in Bars without Chocolate Coating**	**Texture Attributes**			
	**Hardness (N)**	**Fracturability (N)**	**Adhesiveness (J)**	**Cohesiveness**
WPC	34.53 ^f^ ± 0.277	0.30 ^a^ ± 0.024	56.23 ^d^ ± 4.336	0.12 ^d^ ± 0.001
RPC	18.64 ^c^ ± 0.327	0.11 ^a^ ± 0.018	34.16 ^c^ ± 2.166	0.05 ^b^ ± 0.001
SPI	16.38 ^b^ ± 0.201	16.39 ^b^ ± 0.306	5.88 ^a^ ± 0.681	0.03 ^a^ ± 0.002
SUP	149.19 ^h^ ± 0.198	122.52 ^e^ ± 0.439	16.23 ^b^ ± 2.171	0.15 ^e^ ± 0.001
WHP	81.31 ^g^ ± 0.172	0.08 ^a^ ± 0.004	130.15 ^e^ ± 2.157	0.22 ^g^ ± 0.010
HEP	21.50 ^e^ ± 0.170	35.59 ^c^ ± 0.450	3.37 ^a^ ± 0.479	0.09 ^c^ ± 0.006
PAP	13.62 ^a^ ± 0.246	36.81 ^d^ ± 0.217	1.69 ^a^ ± 0.246	0.06 ^b^ ± 0.004
PMP	19.67 ^d^ ± 0.167	0.03 ^a^ ± 0.004	322.85 ^f^ ± 2.695	0.21 ^g^ ± 0.006
ALP	288.50 ^i^ ± 0.326	0.07 ^a^ ± 0.004	27.79 ^c^ ± 2.947	0.17 ^f^ ± 0.002

**Table 3 foods-09-01467-t003:** Effect of different protein sources on high-protein bars cutting resistance.

Type of Protein Used in Bars with Chocolate Coating	Cutting Resistance	Type of Protein Used in Bars without Chocolate Coating	Cutting Resistance
	Force (N)		Force (N)
WPC	79.31 ^f^ ± 0.298	WPC	128.39 ^h^ ± 0.393
RPC	25.75 ^d^ ± 0.198	RPC	25.36 ^e^ ± 0.084
SPI	109.69 ^g^ ± 0.112	SPI	98.21 ^g^ ± 0.162
SUP	22.35 ^c^ ± 0.298	SUP	14.38 ^d^ ± 0.149
WHP	10.54 ^a^ ± 0.073	WHP	8.43 ^b^ ± 0.020
HEP	15.59 ^b^ ± 0.271	HEP	10.58 ^c^ ± 0.126
PAP	75.34 ^e^ ± 0.222	PAP	81.45 ^f^ ± 0.194
PMP	22.70 ^c^ ± 0.143	PMP	7.69 ^a^ ± 0.148
ALP	235.45 ^h^ ± 0.366	ALP	166.82 ^i^ ± 0.138

Data are presented as means ± SD (standard deviation). ^a–i^ Means in the same column with different superscripts are significantly different (*p* < 0.05, Tukey’s HSD test).

**Table 4 foods-09-01467-t004:** (**a**,**b**) Effect of different protein application on color of high-protein bars without chocolate coating measured with Computer Vision System (CVS).

(**a**)				
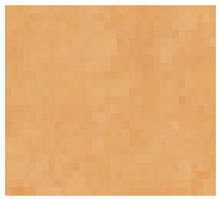 WPC	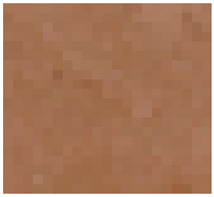 RPC		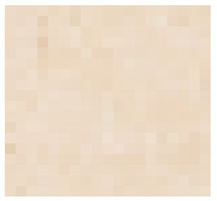 SPI	
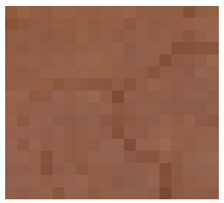 SUP	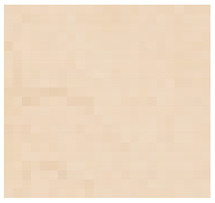 PAP		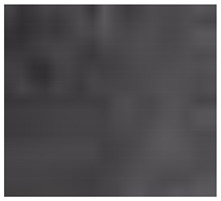 HEP	
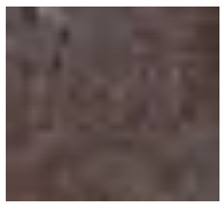 PMP	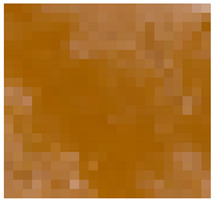 ALP		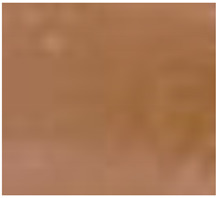 WHP	
(**b**)				
**Type of Protein Used in Bars without Chocolate Coating**	**Attributes**			
	**L***	**a***	**b***	**NBS Units**
WPC	63.86 ^f^ ± 1.069	17.29 ^d^ ± 0.488	44.43 ^f^ ± 0.976	-
RPC	43.71 ^c^ ± 0.488	10.00 ^c^ ± 0.000	21.14 ^b^ ± 0.690	29.12
SPI	79.43 ^h^ ± 0.787	5.29 ^a^ ± 0.488	19.43 ^a^ ± 0.976	29.26
SUP	34.14 ^a^ ± 0.378	11.00 ^c^ ± 0.000	22.29 ^b^ ± 0.756	34.58
WHP	40.57 ^b^ ± 0.976	10.00 ^c^ ± 0.000	31.71 ^c^ ± 1.113	25.32
HEP	50.57 ^e^ ± 0.976	47.43 ^f^ ± 1.272	50.29 ^g^ ± 1.704	30.78
PAP	78.00 ^g^ ± 0.000	8.00 ^b^ ± 0.000	23.00 ^b^ ± 0.000	25.12
PMP	49.43 ^e^ ± 1.134	40.71 ^e^ ± 1.113	34.14 ^d^ ± 2.116	27.02
ALP	47.71 ^d^ ± 0.488	10.86 ^c^ ± 0.378	39.71 ^e^ ± 0.488	16.57

Data are presented as means ± SD (standard deviation). ^a–h^ Means in the same column with different superscripts are significantly different (*p* < 0.05, Tukey’s HSD test). “-“ NBS Units of WPC protein-reference sample, not subject to calculation.

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
