# Peer review of "The Effect of Protein Source on the Physicochemical, Nutritional Properties and Microstructure of High-Protein Bars Intended for Physically Active People"

_foods, 2020, doi:10.3390/foods9101467_

Round 1
Reviewer 1 Report
Dear Authors,
The manuscript is interesting, well-prepared and with appropriate discussion. I would like to suggest some changes.
Line 19: hemp, pea, and whey) …
Line 47: please replace “are trying to find solutions that allow for the substitution of” by “searching for suitable alternatives to”
Lines 53-95: this paragraph is excessively long; please finish the paragraphs in the lines 63, 69, 78, and 88.
Lines 69-71 and 72-74: please include a reference for each of these statements
Line 130: please define the aroma (chocolate, vanilla, …)
Line 218: please remove “such as”
Conclusion: This section is excessively long and must be reduced to a short paragraph of 15-20 lines, maybe. Please remove the information that has already been discussed in previous sections of the Discussion. Anyway, important questions remain: Which protein(s)/isolate(s) do you recommend? Do you recommend the application of chocolate coating in this/these reformulated bar(s)? It is also important to indicate that further studies evaluate the presence of spoilage and pathogenic microorganisms and also characterize and define the shelf life of the bars.
Author Response
University of Life Sciences in Lublin
Faculty of Food Sciences and Biotechnology
Department of Milk Technology and Hydrocolloids
Skromna 8, 20-704 Lublin
Poland
Phone: +48 81 4623350
Fax: +48 81 4623345
E-mail: [email protected]
October 10, 2020,
Dear Reviewer 1,
Our manuscript entitled “The effect of protein source on the physicochemical, nutritional properties and microstructure of high-protein bars intended for physically active people” (Ref. No.: foods-962440 – Round 1) has been revised and is being re-submitted for publication in Foods Special Issue "Sustainable Functional Food Processing".
We have carefully considered each of the comments and made the appropriate revisions in the manuscript. An itemized list of our responses to each of the comments is included below.
Thank you for your kind attention.
Yours faithfully,
Bartosz G. Sołowiej
We have corrected our manuscript with regard to Reviewer 1 comments:
Reviewer #1: yellow color
If both Reviewers had the same comments – pink color
Reviewer #1: Review of Manuscript Number: foods-962440
Title: The effect of protein source on the physicochemical, nutritional properties and microstructure of high-protein bars intended for sportsmen
- Abstract
L 19: (…) hemp, pea, and whey)
Thank you for your comment. We have corrected the content of the parentheses (line: 20).
- Introduction
L 47: please replace “are trying to find solutions that allow for the substitution of” by “searching for suitable alternatives to”
Thank you for your comment. We have replaced the content of the sentence (line: 48).
L 53-95: this paragraph is excessively long; please finish the paragraphs in the lines 63, 69, 78, and 88.
Thank you for your comment. We have divided the content of this part into separate paragraphs (lines: 65, 72, 81 and 92).
L 69-71 and 72-74: please include a reference for each of these statements
Thank you for your comment. In lines 71 and 76 we have added references.
Ermiş, E.; Karasu, E.N. PÜSKÜRTMELİ Kurutucuİle Yaği AlinmiAyçi̇çeği̇Protei̇n Ekstrakti TozÜreti̇mi̇:Fonksi̇yoneÖzelli̇kleri̇ Ve TozKarakteri̇zasyonu. Gida / J. Food 2020, 45, 39–49, doi:10.15237/gida.gd19096.
Pihlanto, A.; Mattila, P.; Mäkinen, S.; Pajari, A.M. Bioactivities of alternative protein sources and their potential health benefits. Food Funct. 2017, doi:10.1039/c7fo00302a.
- Materials and Methods
L 130: please define the aroma (chocolate, vanilla, …)
Thank you for your comment. We have added an information about the ingredient. We have used a natural vanilla aroma in powder (lines: 152 – 153).
Line 218: please remove “such as”
Thank you for your comment. We have corrected the sentence in lines 255-256.
- Conclusion
Conclusion: This section is excessively long and must be reduced to a short paragraph of 15-20 lines, maybe. Please remove the information that has already been discussed in previous sections of the Discussion. Anyway, important questions remain: Which protein(s)/isolate(s) do you recommend? Do you recommend the application of chocolate coating in this/these reformulated bar(s)? It is also important to indicate that further studies evaluate the presence of spoilage and pathogenic microorganisms and also characterize and define the shelf life of the bars.
Thank you for your comment. In lines 645-671 we have provided much shortened version of the “Conclusions” section (shortened from 42 lines to 26), including all the Reviewer’s suggestions.
Thank you very much for all comments.

Reviewer 2 Report
The manuscript titled “The effect of protein source on the physicochemical, nutritional properties and microstructure of high-protein bars intended for sportsmen” deals within the scope of the Foods journal, by investigating an interesting topic and should be of interest to the readers.
However, some small improvements can be done before the publication. Please find below some remarks to help the revision of the manuscript.
Line 4 (Title): I suggest that the authors consider deleting the part “…intended for sportsmen”. Although this product is mostly intended for sportsmen, perhaps this should not be emphasized in the title because this type of product has its place on the market for a much wider population. However, this is not an objection but just a suggestion.
Line 95: The authors should include a paragraph presenting the literature overview of the usage of soy, pea and pumpkin proteins.
Line 130: Please be more specific about the aroma.
Line 155 and 166: The term "puncture" is usually used when the analysis is performed with a puncture probe or needle. It is better to use the term "compressed" because 12 mm diameter sample was compressed with a 36 mm diameter probe.
Line 168: it is necessary to explain how the texture parameters (hardness, fracturability, adhesiveness and cohesiveness) are determined (calculated) from the TPA curve.
Line 203: The methodology for ultrasonic viscosity determination needs to be explained in more detail, especially because the authors argue that this is a new technique in high-protein bar determination.
Line 330 (Table 3): Maybe the better title of this table will be “Effect of different protein sources on high-protein bars cutting resistance.” Furthermore, consider replacing “Texture attributes” with “Cutting resistance”.
Lines 334-339: The authors should refer this paragraph to Table 3.
Line 361: Please check this. Is it Figure 1?
Lines 408-413 (Figures 2a and 2b): I suggest that the order of the samples in Figures be the same so that the reader can more easily compare the data.
Line 439: The authors should delete this part of the sentence “even though it is currently a new and still not very popular method of food color determination” since there is numerous examples of the research papers dealing with the use of CVS and image analyses in color determination of food products.
Lines 475-478 (Figures 3a and 3b): Same as for Figures 2a and 2b
Line 581: ALP instead of ALG.
Line 606 (Conclusions): Conclusions should be significantly shortened. The authors should present only the most significant conclusions, and not repeat the results of the research.
Author Response
University of Life Sciences in Lublin
Faculty of Food Sciences and Biotechnology
Department of Milk Technology and Hydrocolloids
Skromna 8, 20-704 Lublin
Poland
Phone: +48 81 4623350
Fax: +48 81 4623345
E-mail: [email protected]
October 10, 2020,
Dear Reviewer 2,
Our manuscript entitled “The effect of protein source on the physicochemical, nutritional properties and microstructure of high-protein bars intended for physically active people” (Ref. No.: foods-962440 – Round 1) has been revised and is being re-submitted for publication in Foods Special Issue "Sustainable Functional Food Processing".
We have carefully considered each of the comments and made the appropriate revisions in the manuscript. An itemized list of our responses to each of the comments is included below.
Thank you for your kind attention.
Yours faithfully,
Bartosz G. Sołowiej
We have corrected our manuscript with regard to Reviewer 2 comments:
Reviewer #2: green color
If both Reviewers had the same comments – pink color
Reviewer #2: Review of Manuscript Number: foods-962440
Title: The effect of protein source on the physicochemical, nutritional properties and microstructure of high-protein bars intended for sportsmen
- Title
L 4 (Title): I suggest that the authors consider deleting the part “…intended for sportsmen”. Although this product is mostly intended for sportsmen, perhaps this should not be emphasized in the title because this type of product has its place on the market for a much wider population. However, this is not an objection but just a suggestion.
Thank you for your comment. We have replaced word “sportsmen” in the title with “physically active people”. In this case, the target group will be all people who show any physical activity (lines: 4-5), considering that protein bars are designed primarily for people who exercise.
L 95: The authors should include a paragraph presenting the literature overview of the usage of soy, pea and pumpkin proteins.
Thank you for your comment. We have added suggested information in lines 97-114.
New references:
Vinayashree, S.; Vasu, P. Biochemical, nutritional and functional properties of protein isolate and fractions from pumpkin (Cucurbita moschata var. Kashi Harit) seeds. Food Chem. 2020, doi:10.1016/j.foodchem.2020.128177.
Rezig, L.; Riaublanc, A.; Chouaibi, M.; Guéguen, J.; Hamdi, S. Functional Properties of Protein Fractions Obtained from Pumpkin (Cucurbita Maxima) Seed. Int. J. Food Prop. 2016, 19, 172–186, doi:10.1080/10942912.2015.1020433.
Xu, Y.T.; Liu, L.L. Structural and Functional Properties of Soy Protein Isolates Modified by Soy Soluble Polysaccharides. J. Agric. Food Chem. 2016, doi:10.1021/acs.jafc.6b02737.
Lam, A.C.Y.; Can Karaca, A.; Tyler, R.T.; Nickerson, M.T. Pea protein isolates: Structure, extraction, and functionality. Food Rev. Int. 2018.
L 130: Please be more specific about the aroma.
Thank you for your comment. We have added an information about the ingredient. We have used a natural vanilla aroma in powder (lines: 152 – 153).
L 155 and 166: The term "puncture" is usually used when the analysis is performed with a puncture probe or needle. It is better to use the term "compressed" because 12 mm diameter sample was compressed with a 36 mm diameter probe.
Thank you for your comment. We have replaced term ”puncture” with “compressed” in the sentence (line: 190).
L 168: it is necessary to explain how the texture parameters (hardness, fracturability, adhesiveness and cohesiveness) are determined (calculated) from the TPA curve.
Thank you for your comment. We have added a fragment in which we have included an explanation of the description of TPA parameters (lines: 193-198).
L 203: The methodology for ultrasonic viscosity determination needs to be explained in more detail, especially because the authors argue that this is a new technique in high-protein bar determination.
Thank you for your comment. We have added a fragment in which we have included an explanation of the description of ultrasonic viscosity (lines: 238-246).
New references:
Balter, A.; Szubiakowski, J. Fluorescence probes of viscosity: A comparative study of the fluorescence anisotropy decay of perylene and 3,9-dibromoperylene in glycerol. J. Fluoresc. 1993, doi:10.1007/BF00865272.
Galant, S. Formation of collagen-like structure of gel after radiation modification of gelatin. Colloid Polym. Sci. 1983, doi:10.1007/BF01418221.
Sheen, S.H.; Chien, H.-T.; Raptis, A.C. An in-Line Ultrasonic Viscometer. In Review of Progress in Quantitative Nondestructive Evaluation; 1995.
L 330 (Table 3): Maybe the better title of this table will be “Effect of different protein sources on high-protein bars cutting resistance.” Furthermore, consider replacing “Texture attributes” with “Cutting resistance”.
Thank you for your comment. We have changed the Table 3 title and descriptions of its content (line: 367).
L 334-339: The authors should refer this paragraph to Table 3.
Thank you for your comment. We have added an information about the reference to Table 3 (line: 375).
L 361: Please check this. Is it Figure 1?
Thank you for your comment. We have checked this. Figure 1 is correct. We have decided to present the microstructure photos as 9 photos marked with letters "a-i" to save as much space as possible (lines: 396-398).
L 408-413 (Figures 2a and 2b): I suggest that the order of the samples in Figures be the same so that the reader can more easily compare the data.
Thank you for your comment. In both Figures: 2a and 2b we have given the data in the same order for easier comparison (lines: 446-455).
L 439: The authors should delete this part of the sentence “even though it is currently a new and still not very popular method of food color determination” since there is numerous examples of the research papers dealing with the use of CVS and image analyses in color determination of food products.
Thank you for your comment. We have removed the above-mentioned fragment from the sentence (lines: 476-478).
L 475-478 (Figures 3a and 3b): Same as for Figures 2a and 2b
Thank you for your comment. In both Figures: 3a and 3b we have given the data in the same order for easier comparison (lines: 513-522).
L 581: ALP instead of ALG
Thank you for your comment. We have corrected the nomenclature error (line: 620).
L 606 (Conclusions): Conclusions should be significantly shortened. The authors should present only the most significant conclusions, and not repeat the results of the research.
Thank you for your comment. In lines 645-671 we have provided much shortened version of the “Conclusions” section (shortened from 42 lines to 26), including all the Reviewer’s suggestions.
Thank you very much for all comments.
